# Psychological impact, coping behaviors, and traumatic stress among healthcare workers during COVID-19 in Taiwan: An early stage experience

Meng-Chun Lee[1,2‡], Cheng-Hsu Chen[3‡], Pei-Hsuan Hsieh[1,2], Cheng-Hua Ling[1,2], Cheng-Chia Yang[4], Yu-Chia Chang[5], Li-Yeuh Yeh[1], Hung-Chang Hung[6☯ *], Te-Feng Yeh [2☯ *]

1 Department of Nursing, Nantou Hospital, Ministry of Health and Welfare, Nantou, Taiwan, ROC,
2 Department of Healthcare Administration, Central Taiwan University of Science and Technology, Taichung, Taiwan, ROC, 3 Department of Nephrology, Taichung Veterans General Hospital, Taichung, Taiwan, ROC,
4 Department of Healthcare Administration, Asia University, Taichung, Taiwan, ROC, 5 Department of Long-Term Care, National Quemoy University, Kinmen, Taiwan, ROC, 6 President Office, Nantou Hospital, Ministry of Health and Welfare, Nantou, Taiwan, ROC

☯ These authors contributed equally to this work.
‡ M-CL and C-HC also contributed equally to this work and also share first authorship on this work.
* tfyeh@ctust.edu.tw (T-FY); h550327@yahoo.com.tw (H-CH)

**Data Availability Statement:** All relevant data are within the manuscript and its Supporting Information files.

## Abstract

### Objective

This study investigated the psychological impact on, coping behaviors of, and traumatic stress experienced by healthcare workers during the early stage of the COVID-19 pandemic and formulated effective support strategies that can be implemented by hospitals and government policymakers to help healthcare staff overcome the pandemic.

### Methods

This cross-sectional study recruited clinical healthcare workers at a regional hospital in Nantou County, Taiwan. The questionnaire collected personal characteristics, data on the impact and coping behaviors of the pandemic, and Impact of Event Scale-Revised (IES-R). A total of 354 valid questionnaires were collected. The statistical methods employed were univariate and multivariate stepwise regression, and logistic regression.

### Results

Perceived impact and coping behaviors were found to be moderate in degree, and traumatic stress was lower than that in other countries. However, our data identified the following subgroups that require special attention: those with young age, those living with minor children, nurses, those with self-rated poor mental health, and those with insufficient COVID-19-related training.

**Funding:** The funders had no role in study design, data collection and analysis, decision to publish, or preparation of the manuscript.

**Competing interests:** The authors have declared that no competing interests exist.

## Conclusion

Managers should pay particular attention to helping healthcare workers in high-risk groups, strengthen COVID-19 training, provide adequate protective equipment and shelter, and offer psychological counseling.

## Introduction

In December 2019, the first COVID-19 case was reported in the city of Wuhan, province of Hubei, China. The virus then spread quickly not only to the whole of China but also across the world, resulting in many infections and deaths, including those of numerous healthcare workers (HCWs) who were infected while caring for patients. The World Health Organization (WHO) declared COVID-19 a pandemic on March 11, 2020 [1]. Taiwan is located approximately 100 miles off the coast of southeastern China, and many people frequently travel between Taiwan and China. Therefore, Taiwan was predicted to have the second highest number of confirmed COVID-19 cases [2]. In addition, given that Taiwan was one of the worst-hit territories in the severe acute respiratory syndrome (SARS) outbreak of 2003 [3], people in Taiwan, including HCWs, were anxious about the spread of COVID-19 into Taiwan.

Although most people have been able to stay at home to minimize COVID-19 transmission, HCWs have had to go to clinics and come into contact with (potentially) infected patients, thus placing themselves at high infection risk. This has caused HCWs to experience physical and mental exhaustion, difficult triage decisions, the pain of losing patients and colleagues, and high risk of infection due to the high pressure of caring for patients and the infection of colleagues [4].

As the world has experienced multiple cycles of COVID-19 outbreaks and recovery, many studies have evaluated the impact of the pandemic on the mental health of HCWs [5]. Marvaldi, Mallet, Dubertret, Moro, and Guessoum (2021) indicated that during the COVID-19 pandemic, HCWs have been exposed to causes of potential trauma and stress, including unpredictability of the daily caseload, frequent management of patients and their families' expectations in unexpected situations, burdens of decision-making, high daily fatality rates, and frequent updates to hospital procedures [6]. Major and Hlubocky (2021) in their review indicated the following mental health problems among HCWs during COVID-19: anxiety (7.0%–97.3%), depression (10.6%–62.1%), stress (2.2%–93.8%), posttraumatic stress (3.8%–56.6%), insomnia (8.3%–88.4%), and burnout (21.8%–46.3%) [5].

Lai et al. (2020) indicated more stress among HCWs in Wuhan compared with those outside Wuhan [7]. In most of 2020, compared with other countries suffering from the impact of the COVID-19 pandemic, Taiwan had few imported and locally transmitted cases. Lai et al. (2021) described the policy of the Taiwanese government in response to the COVID-19 pandemic in the first half of 2020, which is exactly the context of this study. Based on Taiwan's experience of the SARS and H1N1 outbreaks, the Taiwanese government constructed a new pandemic prevention strategy and established a public health response mechanism that enable quick action to be taken against future crises. On January 20, 2020, the Central Epidemic Command Center (CECC) was established to coordinate interministerial responses and to integrate the coping policy to prevent a pandemic in Taiwan. The early response strategies included strengthening border control, using the cloud to build a person's travel history for the information of physicians, collecting and distributing personal protective equipment and personal protective equipment for HCWs, and restricting overseas travel plans for HCWs. All hospitals

were required to store sufficient personal protective equipment and close most entrances to facilitate the checking of visitors' TOCC (travel history, occupation, contact history, and cluster) and temperature; all employees and visitors had to wear masks, and outdoor screening stations had to be established to prevent nosocomial infections. From April 12, 2020, the absence of locally transmitted cases enabled the CECC to begin lifting the epidemic prevention measures gradually to allow the public to engage in some normal activities. Although students' winter vacation was postponed due to the pandemic and schools were preparing for distance teaching, the entire semester passed smoothly. When the pandemic control requirements were met, all citizens could feel safe to shop and dine at any store or restaurant [2]. The present study assessed the psychological impact on and coping behaviors of HCWs during the early stage of the COVID-19 pandemic in Taiwan.

## Materials and methods

### Study setting and approval

This study followed the regulation of the Human Subjects Research Act in Taiwan. The study protocol was approved by the Institutional Review Board of Taso-Tun Psychiatric Center, Ministry of Health and Welfare (Protocol no./IRB No: 109032). Verbal informed consent was provided by survey participants (all HCWs in study hospital were included) prior to their enrollment in their working units. The participants were allowed to terminate the survey at any time they desired. The completed questionnaires were placed in sealed envelopes and returned by the researcher to each work unit. The survey was anonymous, and confidentiality of information was assured.

The study was a cross-sectional, hospital-based survey conducted through census from July 15, 2020, to July 3, 2021. The study hospital was a 387-bed government-owned teaching hospital that is also the major government-designated hospital in Nantou County. Of 64 physicians and 314 nurses, valid responses were obtained from 55 physicians and 299 nurses (n = 354, valid response rate: 93.7%).

### Measures

The survey questionnaire addressed personal characteristics, support environment, perceived impact and coping behaviors of COVID-19, and the Impact of Event Scale Revised (IES-R). Researchers have developed items related to support environment, the perceived impact, and coping behaviors of COVID-19 [8–15].

**1. Personal characteristics.** The personal characteristics included the participants' sociodemographic and work-related characteristics. The sociodemographic characteristics were age; gender; having a spouse or partner; having dependent children; living alone; living with a spouse or partner, parents or in-laws, minor children, or adult children; education level; self-rated physical health; self-rated mental health; having quarantine relatives and friends; and household income vulnerability.

The work-related characteristics were occupation (physician or nurse), supervisor position, years of clinical experience, clinical experience during SARS, contact with or caring for patients with confirmed or suspected COVID-19 patients, and participation in COVID-19-related training.

**2. Support environment.** Support environment was divided into hospital support (six items), family and colleague support (two items), and government policy support (two items). All items were rated on a 5-point Likert scale (0 = never, 1 = almost never, 2 = sometimes, 3 = fairly often, 4 = always). The overall Cronbach's α was 0.903, and that of the three domains ranged from 0.582 to 0.875.

**3. Perceived impact of COVID-19.**   Perceived impact of COVID-19 was measured using a 17-item scale evaluating social and psychological stressors, which was developed on the basis of the literature. All items were rated on a 5-point Likert scale (0 = never, 1 = almost never, 2 = sometimes, 3 = fairly often, 4 = always), with higher mean scores indicating a greater perceived impact of the COVID-19 pandemic. Exploratory factor analysis on the 17 items assessing perceived impact of COVID-19 yielded four factors (explaining 65.5% of the total variance; Kaiser–Meyer–Olkin = 0.905): increased work stress, worry about an uncontrollable pandemic, less frequent social activities, and social isolation. The overall Cronbach's α was 0.889, and that of the four domains ranged from 0.552 to 0.893.

**4. Coping behaviors.**   In this study, 20 items were developed for measuring the possible coping behaviors of HCWs, with the items based on those in prior studies. All items were rated on a 5-point Likert scale (0 = never, 1 = almost never, 2 = sometimes, 3 = fairly often, 4 = always). Exploratory factor analysis yielded four factors (explaining 67.5% of the total variance; Kaiser–Meyer–Olkin = 0.871): protection measures, exposure reduction, positive mindfulness, and negative avoidance. The overall Cronbach's α was 0.840, and that of the four domains ranged from 0.786 to 0.901.

**5. The impact event scale-revised, IES-R.**   Cabarkapa et al. (2020) reviewed the literature and concluded that post-trauma stress syndrome is the psychiatric disorder that is most often diagnosed [16]. The IES-R is the most used standardized instrument for measuring subjective distress caused by traumatic events [12]. It is a validated 22-item self-report scale that assesses posttraumatic stress disorder (PTSD), including disorders related to trauma, disturbing memories, and persistent negative emotions resulting from the COVID-19 pandemic [17]. All items were rated on a 5-point Likert scale (0 = not at all, 1 = a little, 2 = moderately, 3 = considerably, 4 = extremely), and the total score was 0–88 points. To identify the risk of PTSD, the total IES-R score was graded for severity. Creamer et al. (2003) proposed that a total score of ≥33 points provides the highest diagnostic accuracy for PTSD [18], whereas Chew et al. (2020), Civantos et al. (2020), and Roberts et al. (2020) used 24 points as the threshold to indicate a clinically significant stress response and define PTSD as a clinical concern [19–21]. To enable comparison with other studies, both thresholds were considered in this study.

## Statistical analysis

Personal characteristics and IES-R thresholds (≥24 and ≥33) are reported as the number and percentage, whereas support environment, perceived impact, coping behaviors, and the overall IES-R score are reported as the mean ± standard deviation. Univariate and multivariate stepwise regression and logistic regression were used to identify the personal characteristics associated with support environment, perceived impact, coping behaviors, and IES-R score. Univariate and multivariate regression and logistic regression were also used to illustrate the associations of support environment, perceived impact, and coping behaviors with IES-R score after adjustment for personal characteristics.

## Results

### Personal characteristics and support environment

The participants' age and years of clinical experience were 37.3 ± 9.0 and 14.1 ± 8.9 years, respectively. Most participants were female (82.8%); the largest age group was 30–39 years (42.4%); most participants had a spouse or partner (59.6%), dependent children (51.7%), and an undergraduate educational level (55.9%). Moreover, 55.9% lived with their spouse or partner, 53.4% lived with their parents or in-laws, 39.0% lived with their minor children, 6.8% lived with their adult children, and only 3.4% lived alone. In all, 68.1% and 69.2% of the

**Table 1. Sociodemographic and work-related characteristics of responders.**

| Personal characteristics | NO. | % | Personal characteristics | NO. | % |
|---|---|---|---|---|---|
| **Total** | **354** | | | | |
| **Gender** | | | **Self-rated mental health** | | |
| male | 61 | 17.2% | Good | 245 | 69.2% |
| female | 293 | 82.8% | Poor | 109 | 30.8% |
| **Age** | | | **Quarantined relatives and friends** | | |
| Under 29 | 74 | 20.9% | Yes | 35 | 9.9% |
| 30–39 years old | 150 | 42.4% | No | 319 | 90.1% |
| 40–49 years old | 94 | 26.5% | **Household income vulnerability** | | |
| Over 50 years old | 36 | 10.2% | Yes | 72 | 20.3% |
| **Spouse or partner** | | | No | 282 | 79.7% |
| Yes | 211 | 59.6% | **Occupation** | | |
| No | 143 | 40.4% | Physician | 55 | 15.5% |
| **Dependent children** | | | Nurse | 299 | 84.5% |
| Yes | 183 | 51.7% | **Supervisor position** | | |
| No | 171 | 48.3% | Yes | 37 | 10.5% |
| **Living alone** | | | No | 317 | 89.5% |
| Yes | 12 | 3.4% | **Years of clinical experience** | | |
| No | 342 | 96.6% | Within 5 years | 56 | 15.8% |
| **Living with[a]** | | | 5–14 years | 131 | 37.0% |
| Spouse or partner | 198 | 55.9% | 15–24 years | 120 | 33.9% |
| Parents or in-laws | 189 | 53.4% | More than 25 years | 47 | 13.3% |
| Minor children | 138 | 39.0% | **Clinical experience during SARS** | | |
| Adult children | 24 | 6.8% | Yes | 115 | 32.5% |
| **Education level** | | | No | 239 | 67.5% |
| College and below | 96 | 27.1% | **Contact with COVID-19 patients** | | |
| Undergraduate | 198 | 55.9% | Yes | 213 | 60.2% |
| Graduate | 60 | 17.0% | No | 141 | 39.8% |
| **Self-rated physical health** | | | **Participation in COVID-19-related training** | | |
| Good | 241 | 68.1% | Yes | 249 | 70.3% |
| Poor | 113 | 31.9% | No | 105 | 29.7% |

a: This question group is multiple choice, each question is with answered yes or no.

respondents rated their physical and mental health, respectively, as good or very good. Approximately a tenth (9.9%) of the respondents had a relative or friend who was quarantined, and the household income of 20.3% was affected by the COVID-19 pandemic (Table 1).

Regarding work-related characteristics, 84.5% were nurses and 10.5% were supervisors; 15–24 years (33.9%) was the most common number of years of clinical experience; 32.5% of the participants were engaged in clinical work during the SARS outbreak; 60.2% had been in contact with or cared for patients with confirmed or suspected COVID-19; and 70.3% had received COVID-19-related training (Table 1).

Regarding support environment, family and colleague support (2.91) was the domain with the highest mean score, and the individual scores of the family and colleagues items were similar. The domain with the second highest score was hospital support (2.70). The items with higher mean scores were the provision of adequate protective equipment and sufficient COVID-19 training, whereas the item with a lower score was the provision of accommodation and food services to reduce the risk of transmission to family members. Government policy

support (2.68) had the lowest mean score. The participants agreed that the government had effectively controlled the COVID-19 pandemic (2.93), but the agreement with the statement that reasonable compensation (2.42) had been given to HCWs caring for patients with COVID-19 was significantly lower (Table 2).

## Perceived impact of COVID-19

The highest mean score for perceived impact of COVID-19 among HCWs was for less frequent social activities (3.04), followed by worry about an uncontrollable pandemic (2.70), increased work stress (2.63), and social isolation (1.73). Items with higher mean scores were the following: "I have canceled travel plans and reduced my travel to avoid infection" (3.26), "Wearing protective equipment for a long time incurs physical and mental burdens" (2.91), "I worry about spreading COVID-19 to family members, relatives, and friends" (2.90), "I have reduced my contact with family members or relatives to avoid infection" (2.83), "I worry about my family members being infected with COVID-19" (2.82), "I worry about the lack of COVID-19 care experience and training" (2.74), and "I worry about being infected with COVID-19" (2.73). Significantly lower mean scores were obtained for social isolation and stigma (1.77), and the score for quitting the current job during the COVID-19 pandemic was only 1.30.

The results of univariate and multivariate stepwise regression are presented in Table 3. Univariate regression analysis revealed that women were significantly more worried about an uncontrollable pandemic, less frequent social activities, and social isolation. Furthermore, the participants older than 50 years and those with self-rated good physical health reported significantly lower social isolation scores, whereas those with any of the following characteristics reported significantly higher social isolation scores: no spouse or partner, no dependent children, not living with a spouse or partner, not a supervisor, or a college degree or lower education level. The participants who lived with minor children were significantly more worried about an uncontrollable pandemic, reported significantly less frequent social activities, but reported significantly lower social isolation. The participants with self-rated good physical or mental health had significantly lower increased work stress and worry about an uncontrollable pandemic. Regarding worry about an uncontrollable pandemic and social isolation, nurses' scores were significantly higher than physicians' scores. HCWs with 5–25 years of clinical practice experience were more worried about an uncontrollable pandemic, reported more less frequent social activities; junior HCWs reported a stronger impact of social alienation. Those who had been engaged in clinical work during the SARS outbreak and who had received COVID-19 related training reported significantly lower social isolation; those who came into contact with or cared for patients with confirmed or suspected COVID-19 were significantly more worried about an uncontrollable pandemic.

After adjustment for personal characteristics, multivariate stepwise regression indicated that HCWs over 50 years old reported significantly lower scores for less frequent social activities; those with a spouse or partner reported significantly higher scores for less frequent social activities but significantly lower scores for social isolation; those living with minor children reported significantly higher scores for worry about an uncontrollable pandemic; those with self-rated good physical health reported significantly lower scores for increased work stress, worry about an uncontrollable pandemic, and social isolation; nurses reported significantly more worry about an uncontrollable pandemic and higher social alienation than physicians; those with a clinical practice experience of 15–25 years reported significantly higher worry about uncontrollable pandemic; those who had received COVID-19-related training reported significantly lower scores for social isolation (Table 3).

**Table 2.** The Descriptive statistics of support environment, perceived impact of COVID-19, coping behaviors.

| Items | Mean | SD |
|---|---|---|
| **Support environment** | | |
| **Hospital support** | **2.70** | **0.63** |
| The hospital provides adequate protective equipment. | 2.87 | 0.74 |
| The hospital provides sufficient COVID19 training. | 2.79 | 0.76 |
| The hospital provides adequate time off and a reasonable shift system. | 2.73 | 0.77 |
| The hospital provides clear infection control and protection guidelines. | 2.70 | 0.86 |
| The hospital provides sufficient mental health and psychological stress relief services. | 2.67 | 0.76 |
| Provide accommodation and food services. | 2.45 | 0.90 |
| **Family and colleagues support** | **2.91** | **0.61** |
| Family support can help relieve stress. | 2.93 | 0.65 |
| Encouragement and support among colleagues can help relieve stress. | 2.88 | 0.67 |
| **Government policy support** | **2.68** | **0.72** |
| The government has effectively controlled the pandemic, and the pressure on HCWs has been relieved. | 2.93 | 0.69 |
| The government provides reasonable compensation to HCWs caring for COVID-19 patients. | 2.42 | 0.99 |
| **Perceived impact of COVID-19** | | |
| **Increased work stress** | **2.63** | **0.73** |
| Wearing protective equipment for a long time incurs physical and mental burdens | 2.91 | 0.85 |
| I worry about the lack of COVID-19 care experience and training. | 2.74 | 0.88 |
| The deployment of manpower for other pandemic prevention measures result in a more serious shortage. | 2.68 | 0.89 |
| I often fall into conflicts between professional ethic duty and self-protection during the COVID-19 pandemic. | 2.54 | 0.93 |
| Precautionary measures create impediment to doing job and reduce the quality of care. | 2.29 | 1.11 |
| **Worry about an uncontrollable pandemic** | **2.70** | **0.74** |
| I worry about spreading COVID-19 to family members, relatives, and friends | 2.90 | 0.91 |
| I worry about my family members being infected with COVID-19. | 2.82 | 0.90 |
| I worry about being infected with COVID-19. | 2.73 | 0.97 |
| I worry that when schools or care institutions are closed due to the pandemic, the responsibility of caring for the family must increase. | 2.71 | 0.92 |
| The continuous reports of the COVID-19 pandemic in the media make people feel nervous. | 2.66 | 0.91 |
| I worry about the uncertainty of when the pandemic will be contained | 2.61 | 0.99 |
| I worry about the increase in work caused by the pandemic, and the inability to take care of the family. | 2.50 | 1.02 |
| **Less frequent social activities** | **3.04** | **0.64** |
| I have canceled travel plans and reduced my travel to avoid infection | 3.26 | 0.67 |
| I reduce contact with family members or relatives to avoid infection. | 2.83 | 0.85 |
| **Social isolation** | **1.73** | **0.76** |
| I don't talk about work to avoid worry for my family. | 2.11 | 1.02 |
| I or my family members have been isolated and stigmatized by the community or society because of my work in the hospital. | 1.77 | 1.05 |
| I once considered to quit my current job during the COVID-19 pandemic. | 1.30 | 1.03 |
| **Coping behaviors** | | |
| **Protection measures** | **3.01** | **0.76** |
| I adhere to the protective measure guidelines established by the hospital | 3.24 | 0.85 |
| I follow strict personal protective measures | 3.00 | 0.98 |
| I feel Encourage and support among colleagues. | 2.91 | 0.86 |
| I have received training and education around COVID-19 (including symptoms, transmission routes, treatment, etc.) | 2.89 | 0.87 |
| **Exposure reduction** | **2.79** | **0.95** |

*(Continued)*

**Table 2.** (Continued)

| Items | Mean | SD |
|---|---|---|
| I maintain a proper social distance from others | 2.85 | 0.99 |
| I avoid public transportation. | 2.78 | 1.08 |
| I avoid going out in public places to minimize exposure. | 2.75 | 1.04 |
| **Positive mindfulness** | **2.42** | **0.74** |
| I switch thoughts and facing the pandemic with positive attitude. | 2.77 | 0.90 |
| I chat with family and friends to relieve stress and obtain support | 2.75 | 0.91 |
| I seek sufficient time off and rest to reduce overtime work. | 2.62 | 0.97 |
| I engage in health-promoting behaviors (more rest, more exercise, balanced diet, etc.). | 2.60 | 0.96 |
| I engage in recreational activities (such as sports, reading, listening to music, going to movies, gardening. . . etc.). | 2.58 | 0.92 |
| I practice self-relaxation methods (such as: abdominal breathing, meditation, yoga. . . etc.). | 2.13 | 1.08 |
| I try to find comfort in my religion or spiritual beliefs. | 1.48 | 1.13 |
| **Negative avoidance** | **1.20** | **0.75** |
| I avoid close contact with family members to reduce the risk of infection. | 1.79 | 1.20 |
| I go with the flow, do nothing, and passively accept the risk of a pandemic. | 1.41 | 1.09 |
| I distract attention from the COVID-19 pandemic by staying busy | 1.23 | 1.08 |
| I limit self to receive too much information about COVID-19. | 1.08 | 1.03 |
| I vent emotions by crying, screaming, smashing things, and so on | 1.05 | 1.05 |
| I use cigarettes, alcohol, or drugs to relieve stress. | 0.61 | 1.03 |

## Coping behaviors

According to Table 2, protection measures (3.01) was the most common coping behavior among the HCWs, followed by reduced exposure (2.79), positive mindfulness (2.42), and negative avoidance (1.20). The items with high scores included "I adhere to the protective measure guidelines established by the hospital" (3.24), "I follow strict personal protective measures" (3.00), and "I have received training and education around COVID-19 (including symptoms, transmission routes, treatment, etc.)" (2.89). "I maintain a proper social distance from others" (2.85). "I vent my emotions by crying, screaming, smashing things, and so on" (1.05), and "I use cigarettes, alcohol, or drugs to relieve stress" (0.61) received the lowest scores. In addition, "I practice self-relaxation methods (abdominal breathing, meditation, yoga etc.)" (2.13) and "I try to find comfort in my religion or spiritual beliefs" (1.48) also received lower scores in the positive mindfulness domain.

Univariate regression analysis revealed that the higher the participant's age, the higher their scores for protection measures and positive mindfulness, and the lower their scores for negative avoidance. HCWs with a spouse or partner reported significantly higher scores for protection measures and exposure reduction. Those with dependent children reported significantly higher scores for exposure reduction. Those living with a spouse or partner, minor children, or adult children reported significantly higher scores for protection measures; those who lived with minor children reported significantly higher scores for exposure reduction; and those who lived with adult children reported significantly higher scores for positive mindfulness. Those with a graduate degree reported significantly higher scores for protection measures and significantly lower scores for negative avoidance. Those who had self-rated good physical and mental health reported significantly higher scores for protection measures and positive mindfulness and significantly lower scores for negative avoidance. Those with a quarantined relative or friend reported significantly lower scores for negative avoidance. Compared with nurses, physicians reported significantly higher scores for protection measures but significantly lower

**Table 3. Univariate and multivariate stepwise regression analysis of perceived impact of covid-19 related factors.**

| Personal characteristics | Univariate | | | | Multivariate | | | |
|---|---|---|---|---|---|---|---|---|
| | Increased work stress | Worry about an uncontrollable pandemic | Less frequent social activities | Social isolation | Increased work stress | Worry about an uncontrollable pandemic | Less frequent social activities | Social isolation |
| | B | B | B | B | B | B | B | B |
| **Gender** | | | | | | | | |
| male | -0.186 | -0.239* | -0.182* | -0.298** | | | | |
| female# | | | | | | | | |
| **Age** | | | | | | | | |
| Under 29# | | | | | | | | |
| 30–39 years old | 0.001 | 0.190 | 0.141 | -0.069 | | | | |
| 40–49 years old | -0.080 | 0.029 | 0.174 | -0.142 | | | | |
| Over 50 years old | -0.145 | -0.106 | -0.209 | -0.380* | | | -0.373** | |
| **Spouse or partner** | | | | | | | | |
| Yes | -0.002 | 0.100 | 0.114 | -0.258** | | | 0.156* | -0.191* |
| No# | | | | | | | | |
| **Dependent children** | | | | | | | | |
| Yes | -0.009 | 0.132 | 0.107 | -0.176* | | | | |
| No# | | | | | | | | |
| **Living with** | | | | | | | | |
| Spouse or partner | | | | | | | | |
| Yes | 0.034 | 0.045 | 0.038 | -0.216** | | | | |
| No# | | | | | | | | |
| Minor children | | | | | | | | |
| Yes | 0.036 | 0.167* | 0.148* | -0.184* | | 0.195* | | |
| No# | | | | | | | | |
| **Education level** | | | | | | | | |
| College and below# | | | | | | | | |
| Undergraduate | -0.034 | -0.059 | 0.014 | -0.208* | | | | |
| Graduate | 0.028 | -0.113 | 0.057 | -0.428** | | | | |
| **Self-rated physical health** | | | | | | | | |
| Good | -0.245** | -0.238** | -0.059 | -0.174* | -0.245** | -0.244*** | | -0.173* |
| Poor# | | | | | | | | |
| **Self-rated mental health** | | | | | | | | |
| Good | -0.228** | -0.235** | -0.049 | -0.132 | | | | |
| Poor# | | | | | | | | |
| **Occupation** | | | | | | | | |
| Physician | -0.157 | -0.255* | -0.073 | -0.431*** | | -0.291** | | -0.329** |
| Nurse# | | | | | | | | |
| **Supervisor position** | | | | | | | | |
| Yes | 0.146 | -0.009 | 0.011 | -0.330* | | | | |
| No# | | | | | | | | |
| **Years of clinical experience** | | | | | | | | |
| Within 5 years# | | | | | | | | |
| 5–14 years | 0.053 | 0.270* | 0.186 | -0.263* | | | | |
| 15–24 years | 0.036 | 0.251* | 0.228* | -0.168 | | 0.159* | | |

(*Continued*)

**Table 3.** (Continued)

| Personal characteristics | Univariate | | | | Multivariate | | | |
|---|---|---|---|---|---|---|---|---|
| | Increased work stress | Worry about an uncontrollable pandemic | Less frequent social activities | Social isolation | Increased work stress | Worry about an uncontrollable pandemic | Less frequent social activities | Social isolation |
| More than 25 years | -0.146 | -0.103 | -0.029 | -0.396** | | | | |
| **Clinical experience during SARS** | | | | | | | | |
| Yes | 0.003 | -0.078 | -0.052 | -0.198* | | | | |
| No# | | | | | | | | |
| **Contact with COVID-19 patients** | | | | | | | | |
| Yes | 0.147 | 0.175* | 0.055 | 0.011 | | | | |
| No# | | | | | | | | |
| **Participation in COVID-19-related training** | | | | | | | | |
| Yes | 0.016 | 0.077 | 0.089 | -0.306** | | | | -0.242** |
| No# | | | | | | | | |
| F | | | | | 8.817*** | 6.185*** | 6.923*** | 8.624*** |
| $R^2$ | | | | | 0.024 | 0.066 | 0.038 | 0.090 |
| Adj. $R^2$ | | | | | 0.022 | 0.055 | 0.032 | 0.080 |

*: p<0.05

**: P<0.01

***: p<0.001.

#: Reference group.

Sociodemographic and work-related characteristics of responders had been adjusted.

scores for negative avoidance. Supervisors reported significantly higher scores for protection measures. The greater the clinical experience, the higher the scores for protection measures and positive mindfulness and the lower those for negative avoidance. Those who had been engaged in clinical work during the SARS outbreak reported significantly higher scores for protection measures and positive mindfulness and significantly lower scores for negative avoidance. Those who had received COVID-19-related training reported significantly higher scores for protection measures (Table 4).

After adjustment for personal characteristics, multivariate stepwise regression indicated that HCWs who lived with minor or adult children reported significantly higher scores for exposure reduction. Those with self-rated good physical health reported significantly higher scores for protection measures; those with self-rated good mental health reported significantly higher scores for positive mindfulness and significantly lower scores for negative avoidance. Nurses reported significantly higher scores for negative avoidance than physicians. Those with more than 25 years of clinical experience had the highest scores for positive mindfulness. Those who had been engaged in clinical work during the SARS outbreak reported significantly higher scores for protection measures and significantly lower scores for negative avoidance. Those who had participated in COVID-19-related training reported significantly higher scores for protection measures (Table 4).

## The IES-R

The respondents had a IES-R score of 17.01 ± 14.16 (0–74). Moreover, 96 of the respondents (27.1%) had a score ≥ 24, indicating that PTSD was a clinical concern, and 63 respondents

**Table 4. Univariate and multivariate stepwise regression analysis of coping behaviors related factors.**

| Personal characteristics | Univariate | | | | Multivariate | | | |
|---|---|---|---|---|---|---|---|---|
| | Protection measures | Exposure reduction | Positive mindfulness | Negative avoidance | Protection measures | Exposure reduction | Positive mindfulness | Negative avoidance |
| | B | B | B | B | B | B | B | B |
| **Age** | | | | | | | | |
| Under 29[#] | | | | | | | | |
| 30–39 years old | 0.118 | 0.172 | 0.151 | -0.122 | | | | |
| 40–49 years old | 0.421*** | 0.327* | 0.240** | -0.258* | | | | |
| Over 50 years old | 0.492** | 0.143 | 0.424*** | -0.367* | | | | |
| **Spouse or partner** | | | | | | | | |
| Yes | 0.229** | 0.256* | 0.084 | -0.136 | | | | |
| No[#] | | | | | | | | |
| **Dependent children** | | | | | | | | |
| Yes | 0.154 | 0.212* | 0.040 | -0.076 | | | | |
| No[#] | | | | | | | | |
| **Living with** | | | | | | | | |
| Spouse or partner | | | | | | | | |
| Yes | 0.168* | 0.196 | 0.008 | -0.094 | | | | |
| No[#] | | | | | | | | |
| Minor children | | | | | | | | |
| Yes | 0.190* | 0.269** | 0.106 | -0.074 | | 0.280** | | |
| No[#] | | | | | | | | |
| Adult children | | | | | | | | |
| Yes | 0.425** | 0.389 | 0.344* | -0.099 | | 0.418* | | |
| No[#] | | | | | | | | |
| **Education level** | | | | | | | | |
| College and below[#] | | | | | | | | |
| Undergraduate | 0.257** | 0.163 | 0.040 | -0.172 | | | | |
| Graduate | 0.438*** | 0.387* | 0.067 | -0.322** | | | | |
| **Self-rated mental health** | | | | | | | | |
| Good | 0.264** | -0.050 | 0.360*** | -0.207* | 0.242** | | | |
| Poor[#] | | | | | | | | |
| **Self-rated mental health** | | | | | | | | |
| Good | 0.271** | 0.022 | 0.336*** | -0.206* | | | 0.322*** | -0.184* |
| Poor[#] | | | | | | | | |
| **Quarantined relatives and friends** | | | | | | | | |
| Yes | 0.188 | -0.179 | -0.093 | -0.281* | | | | |
| No[#] | | | | | | | | |
| **Occupation** | | | | | | | | |
| Physician | 0.306** | 0.220 | -0.029 | -0.286** | | | | -0.239* |
| Nurse[#] | | | | | | | | |
| **Supervisor** | | | | | | | | |
| Yes | 0.397** | 0.234 | 0.134 | -0.229 | | | | |
| No[#] | | | | | | | | |
| **Years of clinical experience** | | | | | | | | |
| Within 5 years[#] | | | | | | | | |
| 5–14 years | 0.223 | 0.187 | 0.166 | -0.317** | | | | |

(*Continued*)

**Table 4.** (Continued)

| Personal characteristics | Univariate | | | | Multivariate | | | |
|---|---|---|---|---|---|---|---|---|
| | Protection measures | Exposure reduction | Positive mindfulness | Negative avoidance | Protection measures | Exposure reduction | Positive mindfulness | Negative avoidance |
| | B | B | B | B | B | B | B | B |
| 15–24 years | 0.284* | 0.312* | 0.231 | -0.344** | | | | |
| More than 25 years | 0.636*** | 0.297 | 0.532*** | -0.525*** | | | 0.292* | |
| post hoc | | | | | | | | |
| **Clinical experience during SARS** | | | | | | | | |
| Yes | 0.372*** | 0.148 | 0.166* | -0.295** | 0.280** | | | -0.249** |
| No[#] | | | | | | | | |
| **Participation in COVID-19-related training** | | | | | | | | |
| Yes | 0.440*** | 0.105 | 0.157 | -0.142 | 0.392*** | | | |
| No[#] | | | | | | | | |
| F | | | | | 17.025*** | 5.724** | 13.077*** | 7.310*** |
| $R^2$ | | | | | 0.127 | 0.032 | 0.069 | 0.059 |
| Adj. $R^2$ | | | | | 0.120 | 0.026 | 0.064 | 0.051 |

*: $p < 0.05$

**: $P < 0.01$

***: $p < 0.001$.

[#]: Reference group.

Sociodemographic and work-related characteristics of responders had been adjusted.

(17.8%) had a score $\geq$ 33, indicating moderate or high risk of PTSD. Univariate regression analysis revealed that self-rated poor physical or mental health and household income vulnerability were risk factors for the total IES-R score. Those who were younger, who had self-rated poor physical and mental health, who had household income vulnerability, and who were not engaged in clinical work during the SARS outbreak had significantly higher risks of traumatic stress (Table 5).

After adjustment for personal characteristics, multivariate stepwise regression indicated that IES-R scores were significantly higher for individuals with self-rated poor mental health and household income vulnerability. Moreover, the risk of IES-R score $\geq$ 24 was significantly higher in the participants with self-rated poor mental health, with household income vulnerability, and who were not engaged in clinical work during the SARS outbreak, whereas the risk of IES-R score $\geq$ 33 was significantly higher in those who were not engaged in clinical work during the SARS outbreak (Table 5).

### Factors associated with IES-R

Univariate regression analysis revealed that IES-R score was positively associated with the scores for all four domains of perceived COVID-19 impact, negatively associated with the scores for protection measures and positive mindfulness within coping behaviors, and positively associated with the score for negative avoidance. The higher the scores in the domains of perceived COVID-19 impact, such as worry about an uncontrollable pandemic and less frequent social activities, the higher the risk of IES-R score $\geq$ 24. Those reporting high scores on protection measures and positive mindfulness were more likely to have a lower risk of IES-R score $\geq$ 24, but negative avoidance increased the risk. Worried about an uncontrollable pandemic and social isolation (within

**Table 5. Univariate and multivariate stepwise regression and logistic regression analysis of IES-R related factors.**

| Personal characteristics | Univariate | | | Multivariate | | |
|---|---|---|---|---|---|---|
| | IES-R (n = 354) | IES-R≥24 (n = 96) | IES-R≥33 (n = 63) | IES-R (n = 354) | IES-R≥24 (n = 96) | IES-R≥33 (n = 63) |
| | B | OR | OR | B | OR | OR |
| **Age** | | | | | | |
| Under 29[#] | | | | | | |
| 30–39 years old | -1.629 | 0.682 | 0.566 | | | |
| 40–49 years old | -3.280 | 0.389** | 0.281** | | | |
| Over 50 years old | -4.810 | 0.329* | 0.139* | | | |
| **Good physical health** | | | | | | |
| Yes | -6.274*** | 0.519** | 0.516* | | | |
| No[#] | | | | | | |
| **Self-rated mental health** | | | | | | |
| Good | -6.934*** | 0.511** | 0.482* | -6.943*** | 0.541* | 0.527* |
| Poor[#] | | | | | | |
| **Household income vulnerability** | | | | | | |
| Yes | 4.227* | 1.852* | 1.429 | 4.248* | 1.944* | |
| No[#] | | | | | | |
| **Occupation** | | | | | | |
| Physician | -2.918 | 0.477 | 0.319* | | | |
| Nurse[#] | | | | | | |
| **Clinical experience during SARS** | | | | | | |
| Yes | -2.646 | 0.489* | 0.291** | | 0.506* | 0.309** |
| No[#] | | | | | | |
| F | | | | 12.559*** | | |
| $R^2$ | | | | 0.067 | | |
| Adj. $R^2$ | | | | 0.061 | | |

*: $p<0.05$

**: $P<0.01$

***: $p<0.001$.

[#]: Reference group.

Sociodemographic and work-related characteristics of responders had been adjusted.

perceived impact of COVID-19) and negative avoidance (within coping behaviors) were the risk factors for IES-R score ≥ 33, whereas protection measures, exposure reduction, and positive mindfulness (within coping behaviors) were protective factors (Table 6).

After adjustment for personal characteristics, multivariate regression analyses indicated that IES-R score was positively associated with scores for worry about an uncontrollable pandemic and negative avoidance and negatively associated with scores for protection measures. Multivariate logistic regression analyses indicated that government policy support and negative avoidance were risk factors for IES-R score ≥ 24. Furthermore, negative avoidance score was positively associated with IES-R score ≥ 33, whereas hospital support score was negatively associated with IES-R score ≥ 33(Table 6).

## Discussion

In the first half of 2020, unlike countries that have been severely hit by the COVID-19 pandemic, Taiwan has consistently maintained a very low number of confirmed cases [2]. The

**Table 6. The association of support environment, perceived impact, coping behaviors, and IES-R.**

| Variables | Univariate | | | Multivariate | | |
|---|---|---|---|---|---|---|
| | IES-R | IES-R≥24 | IES-R≥33 | IES-R | IES-R≥24 | IES-R≥33 |
| | (B) | (OR) | (OR) | (B) | (OR) | (OR) |
| **Support environment** | | | | | | |
| Hospital | -1.339 | 0.696 | 0.669 | -0.815 | 0.540 | 0.402* |
| Family and colleague | -1.013 | 0.699 | 0.755 | -0.495 | 0.622 | 1.022 |
| Government policy | 0.253 | 1.057 | 1.013 | 1.111 | 1.761* | 1.609 |
| **Perceived impact of COVID-19** | | | | | | |
| Increased work stress | 2.805** | 1.324 | 1.124 | -1.212 | 0.728 | 0.630 |
| Worry about an uncontrollable pandemic | 5.173*** | 1.905** | 1.569* | 3.373** | 1.512 | 1.292 |
| Less frequent social activities | 2.942* | 1.593* | 1.333 | 1.664 | 1.890 | 1.919 |
| Social isolation | 5.134*** | 2.213*** | 1.978*** | 1.252 | 1.376 | 1.158 |
| **Coping behaviors** | | | | | | |
| Protection measures | -5.297*** | 0.485*** | 0.423*** | -2.502* | 0.620 | 0.634 |
| Exposure reduction | -1.488 | 0.882 | 0.748* | -0.843 | 0.998 | 0.728 |
| Positive mindfulness | -3.263** | 0.676* | 0.652* | -1.049 | 0.762 | 0.692 |
| Negative avoidance | 7.180*** | 2.755*** | 3.481*** | 5.546*** | 2.474*** | 3.829*** |
| F | | | | 10.205*** | | |
| $R^2$ | | | | 0.281 | | |
| Adj. $R^2$ | | | | 0.253 | | |
| $R^2$ change | | | | 0.214*** | | |

*: $p<0.05$

**: $P<0.01$

***: $p<0.001$.

Sociodemographic and work-related characteristics of responders had been adjusted.

mental health of HCWs in various countries was greatly affected by the severe pandemic [5–7], while HCWs in Taiwan were moderately affected in terms of perceived impact and coping behaviors.

The sources of psychological impacts among HCWs during the COVID-19 pandemic may have been fear of their own infection and that of their family, fear of spreading the virus to family and friends, worry about an uncontrollable pandemic, inadequate precautionary measures, unclear and insufficient protection guidelines and training, social isolation and stigmatization, and lack of support from family, colleagues, the hospital, and government [8,22–26]. In the present study, less frequent social activities had the strongest impact during the early stage of the COVID-19 pandemic. Similar to other studies, this study found that HCWs were worried about an uncontrollable pandemic; spreading COVID-19 to family members, relatives, and friends; and themselves and family members being infected with COVID-19. The low mean score for quitting during the COVID-19 pandemic means that the pandemic was not serious, showing the professional ethics of HCWs. During the Middle East respiratory syndrome(MERS) outbreak in 2014, the mortality rate was 36%–70%. Most HCWs chose to stay and work, believing that this was an inherent professional and ethical obligation [12]; in 2020, during the outbreak of COVID-19, HCWs also thought positively of their work. At this difficult time, only professionally trained HCWs could help people face an emerging infectious disease, highlighting the professional ethics of these workers [27].

In this study, the coping strategies most commonly used by HCWs were the following of strict protection measures, training and education about COVID-19, and maintaining a

proper social distance from others. These results are consistent with those of Cai et al [8] and also indicates that safety from infection was the main concern, as HCWs were most worried that they might infect their family with COVID-19. Low scores were reported for venting emotions; using cigarettes, alcohol, or drugs; and seeking religious or spiritual support. The pandemic was not as serious as in other countries may have been a reason. In the United States, frontline emergency HCWs, where the highest prevalence of COVID-19, identified religion-related coping mechanisms such as praying as some of the most important ways to combat the mental and psychological burden of the pandemic [25]. Babore et al. (2020) argued that for professional HCWs, the exhausting work shifts left little personal time for prayer and religion during the pandemic [28]. In addition, many Taiwanese not being enthusiastic about religious activities (18.8% actively participated in the present study) may have been a major reason why the HCWs did not often seek religious comfort.

The mean IES-R score (17.01 ± 14.16) of HCWs in this study was lower than in other studies [7,29,30]. This may indicate that the pandemic has been less severe than other outbreaks. However, the percentages for possible clinical concern of PTSD and moderate to high risk of PTSD were not low compared with those in other studies [31–33] Thus, even if the pandemic was not severe, most HCWs were not under high traumatic stress. However, certain characteristics of HCWs were at higher risk of traumatic stress which require special attention of hospital administrators.

Younger age, having a spouse or partner, living with underage children, and being a nurse were risk factors for strong impact of COVID-19, whereas self-rated good mental health and having received COVID-19-related training were protective factors. HCWs who lived with minors or adult children, with self-rated good physical health, with self-rated good mental health, with 25 years of clinical experience, who had been engaged in clinical work during the SARS outbreak, and who had received COVID-19-related training tended to employ positive coping strategies, whereas nurses were more likely to adopt negative response strategies. Self-rated poor mental health, household income vulnerability, and not being engaged in clinical work during the SARS outbreak were risk factors for traumatic stress. Several studies have illustrated that factors predisposing a person to psychological distress can be divided into sociodemographic, work-related, and societal characteristics. In terms of sociodemographic characteristics, HCWs who were women, were younger, had dependent children, and had self-reported pre-existing psychological or physical ill health were more vulnerable. The work-related risk factors were being a nurse, less clinical experience, contact with or care for patients with COVID-19, and those who perceived precautionary measures to be unsatisfactory. Societal stigma, lack of knowledge and experience of COVID-19, inadequate staff training, organizational support, and compensation have been identified as societal risk factors for mental disorders [5,13,34–37].

Nurses are predominantly female and have more patient contact than other HCWs. The important role of family in traditional Chinese culture results in female nurses bearing a heavier workload and conflict between work and family. Especially those with dependent children, in addition to worrying about being infected and spreading virus to their family, also worry that no one would take good care of their children if their workload increased [12,33,34]. HCWs who were older or who had greater clinical experience, the perception of being adequately trained, and faith in precautionary measures experienced less stress. A general drop in disease transmission also improved psychological outcomes [5,36,37]. In the early stage of the COVID-19 pandemic, lack of COVID-19 knowledge and memory shock during SARS outbreak resulted in the rate of traumatic stress not being lower than in other countries, even if there were not many confirmed cases in Taiwan. The HCWs with traumatic stress were the younger HCWs who had not experienced SARS. After SARS, the Taiwanese government

established an infectious disease pandemic management system; senior HCWs must now participate in a hospital infection control drill at least once a year, especially regarding the use of personal protective equipment. Therefore, the psychological distress of HCWs who had experienced SARS was lower [38].

In this study, the perceived impact of COVID-19 was positively associated with the risk of traumatic stress, especially in those worried about an uncontrollable pandemic. This is consistent with other studies [30–32,39]. The higher an HCW perceived the impact of COVID-19 to be, the higher the psychological pressure and the higher the risk of traumatic stress. Positive coping behaviors—such as protection measures, exposure reduction, and positive mindfulness—can reduce the risk of traumatic stress, whereas negative coping behaviors can increase this risk. Nie et al. (2020) reported that the use of negative coping strategies was positively associated with more severe PTSD and higher psychological distress in frontline nurses during COVID-19 [30], and Wang et al. (2020) reported that positive coping was linked to less severe PTSD whereas negative coping was positively associated with PTSD severity [40]. Si et al. (2020) indicated that passive coping strategies were positively correlated with PTSD severity, whereas active coping strategies were negatively correlated with it; the authors recommended promotion of active coping styles among HCWs to minimize the impact on mental health [26].

## Conclusion

This study investigated the psychological impact, coping behaviors, and traumatic stress of HCWs. The perceived impact and coping behaviors were found to be moderate in degree, and the traumatic stress experienced was lower than that in other countries. However, our data identified the following subgroups that require special attention: those of younger age, those living with minor children, nurses, those with self-rated poor mental health, and those with insufficient COVID-19-related training. Healthcare managers have a substantial role in protecting the mental health of their staff and helping them overcome the impact of the pandemic. Managers should pay particular attention to HCWs in high-risk groups, such as young and inexperienced nurses, because they experience stronger psychological impacts and more traumatic stress, perceive less social support, use fewer positive coping behaviors, and need more care from others. The risk factors for health problems among HCWs in the COVID-19 pandemic are similar to those during the SARS outbreak. Hospitals must strengthen their COVID-19 training, provide adequate protective equipment, provide shelters to reduce workers' worries about infecting their families, and establish a mechanism for psychological counseling.

The study has some limitations. First, the pandemic situation is constantly changing, and this cross-sectional study analyzed information from one time point during the pandemic, so the findings may not be true for the whole pandemic. Second, compared with that in other countries, the pandemic in Taiwan has been less severe, and the investigated hospital is located in a place with almost no cases of COVID-19. Therefore, our study can only partly reflect the mental health outcome of HCWs. However, our findings provide valuable information for policymakers and mental health professionals worldwide regarding the psychological distress faced by individuals during a pandemic.

## Supporting information

**S1 File.**
(XLS)

## Acknowledgments

The authors are grateful to all of the participants who kindly agreed to participate in this survey.

## Author Contributions

**Conceptualization:** Meng-Chun Lee, Cheng-Hsu Chen, Cheng-Chia Yang, Yu-Chia Chang, Hung-Chang Hung, Te-Feng Yeh.

**Data curation:** Meng-Chun Lee, Pei-Hsuan Hsieh, Cheng-Hua Ling, Li-Yeuh Yeh, Te-Feng Yeh.

**Formal analysis:** Cheng-Chia Yang.

**Investigation:** Meng-Chun Lee, Pei-Hsuan Hsieh, Cheng-Hua Ling.

**Methodology:** Meng-Chun Lee, Cheng-Hsu Chen, Pei-Hsuan Hsieh, Cheng-Hua Ling, Cheng-Chia Yang, Yu-Chia Chang, Hung-Chang Hung, Te-Feng Yeh.

**Project administration:** Pei-Hsuan Hsieh, Cheng-Hua Ling, Li-Yeuh Yeh, Hung-Chang Hung.

**Supervision:** Cheng-Hsu Chen, Yu-Chia Chang, Li-Yeuh Yeh, Hung-Chang Hung.

**Validation:** Cheng-Hsu Chen, Cheng-Chia Yang, Yu-Chia Chang, Li-Yeuh Yeh, Hung-Chang Hung.

**Writing – original draft:** Meng-Chun Lee, Te-Feng Yeh.

**Writing – review & editing:** Cheng-Hsu Chen, Hung-Chang Hung, Te-Feng Yeh.

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
