## [Decision Letter · Decision Letter 0]

18 Jul 2022

PONE-D-21-39757Psychological Impact, Coping Behaviors, and Traumatic Stress among Healthcare Workers during COVID-19 in a Low-Disease-Burden Country: An Early Stage ExperiencePLOS ONE

Dear Dr. Yeh,

Thank you for submitting your manuscript to PLOS ONE. After careful consideration, we feel that it has merit but does not fully meet PLOS ONE’s publication criteria as it currently stands. Therefore, we invite you to submit a revised version of the manuscript that addresses the points raised during the review process.

Please note that we have only been able to secure a single reviewer to assess your manuscript. We are issuing a decision on your manuscript at this point to prevent further delays in the evaluation of your manuscript. Please be aware that the editor who handles your revised manuscript might find it necessary to invite additional reviewers to assess this work once the revised manuscript is submitted. However, we will aim to proceed on the basis of this single review if possible.  Your manuscript has been assessed by an expert reviewer, whose comments are appended below. The reviewer has highlighted concerns about aspects of the methodology and discussion. Please ensure you respond to each point carefully in your response to reviewers document, and modify your manuscript accordingly.

We look forward to receiving your revised manuscript.

Kind regards,

Joseph Donlan

Editorial Office

PLOS ONE

Journal Requirements:

Reviewers' comments:

Reviewer's Responses to Questions

**Comments to the Author**

1. Is the manuscript technically sound, and do the data support the conclusions?

Reviewer #1: Yes

2. Has the statistical analysis been performed appropriately and rigorously? 

Reviewer #1: Yes

3. Have the authors made all data underlying the findings in their manuscript fully available?

Reviewer #1: Yes

4. Is the manuscript presented in an intelligible fashion and written in standard English?

Reviewer #1: Yes

5. Review Comments to the Author

Reviewer #1: Dear Editor,

Thank you for inviting me to review the manuscript entitled “Psychological Impact, Coping Behaviors, and Traumatic Stress among Healthcare Workers during COVID-19 in a Low-Disease-Burden Country: An Early Stage Experience”. This study aimed to investigate the psychological impact on, coping behaviors of, and traumatic stress experienced by healthcare workers during the early stage of the COVID-19 pandemic. I would like to suggest the following points for the improvement of the manuscript.

Comments to authors

Dear authors,

Thank you for the dedication and efforts invested in this research study. I would like to suggest you consider the following points for the improvement of the manuscript.

General comments

The title of the manuscript should be including the study setting and design. The word “Low-Disease-Burden Country” seems to be a general term.

Introduction

The statement “given that Taiwan was one of the worst-hit territories in the severe acute respiratory syndrome (SARS) outbreak of 2003……..” needs a reference.

Methods

Please include the sample size estimation for this study.

The sampling method should be mentioned.

How was the data collected from the respondents?

Results

Line 173: Please check the percentage of the respondents in table 1. The total should be 100% for variables, and some variables have got more than 100% (eg, age), and some got less than 100% (eg, education level).

Line 173: In demographic variables, “living with …..” has 155.1%. Did the multiple answers allow for that variable? If yes, pls indicate in the table footnote.

The majority of the respondents were nurses (84.5%) and 15.5% were physicians. Is there any pre-determined ratio depending on the occupation?

Line 285: “The respondents had a mean IES-R score of 17.01 ± 14.16 (0–74).” It would be better if the authors mentioned about mean, SD, minimum – maximum in the sentence.

Discussion

Line 332: The first paragraph of the discussion focused on the context of the study, the COVID-19 situation, and responses in Taiwan. This paragraph is more suitable in the introduction rather than the discussion. The discussion should start with the overall major findings of the study and compare the similarities and differences with previous literature.

The authors may aware that many studies have been investigated the impact of COVID-19 on mental health, coping strategies, and traumatic stress. I would like to suggest the authors to highlight the unique findings of your study and mention them in the discussion.

Limitations

What was the sampling method to recruit the respondents? Is that method affect on the generalizability of the findings?

I would like to suggest keeping a separate section for the recommendation.

Thank you.

6. PLOS authors have the option to publish the peer review history of their article (what does this mean?). If published, this will include your full peer review and any attached files.

Reviewer #1: **Yes: **Mila Nu Nu Htay

---

## [Author Response · Author response to Decision Letter 0]

23 Aug 2022

Dear reviewer

Thank you for your kind review and constructive recommendations. Our reply is as follows:

General comments

The title of the manuscript should be including the study setting and design. The word “Low-Disease-Burden Country” seems to be a general term

Answer：Thanks for your comment.

In order to make the setting of the study more clear to readers, we decided to modify the title to "Psychological Impact, Coping Behaviors, and Traumatic Stress among Healthcare Workers during COVID-19 in Taiwan: An Early Stage Experience". And describe that the impact of the pandemic during this period was significantly lower in Taiwan than in other countries in the introduction.

Introduction

The statement “given that Taiwan was one of the worst-hit territories in the severe acute respiratory syndrome (SARS) outbreak of 2003……..” needs a reference.

Answer：Thanks for your comment and reminder.

We mark cited references and revise the numbering of the references.(Line 56)

Methods

Please include the sample size estimation for this study.

The sampling method should be mentioned.

How was the data collected from the respondents?

Answer：Thanks for your comment.

The study subjects included all HCWs employed by the study hospitals, as there were no sampling issues to consider. The process of data collection was added to the text.(Line105-108)

Results

Line 173: Please check the percentage of the respondents in table 1. The total should be 100% for variables, and some variables have got more than 100% (eg, age), and some got less than 100% (eg, education level).

Answer：Thanks for your comment.

These results are due to rounding. After reviewing the raw data again, some data are corrected to match the total to 100%.(Line 159, Table 1)

Line 173: In demographic variables, “living with …..” has 155.1%. Did the multiple answers allow for that variable? If yes, pls indicate in the table footnote.

Answer：Thanks for your comment.

Answer yes or no to each question as indicated in the footnotes. (Line 196)

The majority of the respondents were nurses (84.5%) and 15.5% were physicians. Is there any pre-determined ratio depending on the occupation?

Answer：Thanks for your comment.

The study hospital employed 64 (16.9%) physicians and 314 nurses (83.1%)(in Line 113), which was not significantly different from the rate of 55 (15.5%) responding physicians and 299 nurse practitioners (84.5%).

Line 285: “The respondents had a mean IES-R score of 17.01 ± 14.16 (0–74).” It would be better if the authors mentioned about mean, SD, minimum – maximum in the sentence.

Answer：Thanks for your comment.

We remove mean so that this sentence represents all data about IES-R.(Line 307)

Discussion

Line 332: The first paragraph of the discussion focused on the context of the study, the COVID-19 situation, and responses in Taiwan. This paragraph is more suitable in the introduction rather than the discussion. The discussion should start with the overall major findings of the study and compare the similarities and differences with previous literature.

Answer：Thanks for your comment.

We moved the text of the first paragraph to the introduction, as suggested by the reviewers.(Line77-91)

The authors may aware that many studies have been investigated the impact of COVID-19 on mental health, coping strategies, and traumatic stress. I would like to suggest the authors to highlight the unique findings of your study and mention them in the discussion.

Answer：Thanks for your comment.

We partially revised the original first paragraph to show that although the pandemic was less severe, the effects of perceived impact and coping behaviors were moderate.(Line 354-355)

Limitations

What was the sampling method to recruit the respondents? Is that method affect on the generalizability of the findings?

Answer：Thanks for your comment.

Because all HCWs in the study hospitals were included in the study subjects, there was no sampling issue in this study. Generalizability of limitations will still be concentrated in a single study setting that had described in research limitations (in Line 463).

The authors once again thank the reviewers for reading this manuscript in detail and for making constructive recommendations.

---

## [Decision Letter · Decision Letter 1]

10 Oct 2022

Psychological Impact, Coping Behaviors, and Traumatic Stress among Healthcare Workers during COVID-19 in Taiwan: An Early Stage Experience

PONE-D-21-39757R1

Dear Dr. Yeh,

We’re pleased to inform you that your manuscript has been judged scientifically suitable for publication and will be formally accepted for publication once it meets all outstanding technical requirements.

Kind regards,

Mila Nu Nu Htay

Guest Editor

PLOS ONE

Additional Editor Comments (optional):

Reviewers' comments:

Reviewer's Responses to Questions

**Comments to the Author**

1. If the authors have adequately addressed your comments raised in a previous round of review and you feel that this manuscript is now acceptable for publication, you may indicate that here to bypass the “Comments to the Author” section, enter your conflict of interest statement in the “Confidential to Editor” section, and submit your "Accept" recommendation.

Reviewer #2: All comments have been addressed

2. Is the manuscript technically sound, and do the data support the conclusions?

Reviewer #2: Yes

3. Has the statistical analysis been performed appropriately and rigorously? 

Reviewer #2: Yes

4. Have the authors made all data underlying the findings in their manuscript fully available?

Reviewer #2: Yes

5. Is the manuscript presented in an intelligible fashion and written in standard English?

Reviewer #2: Yes

6. Review Comments to the Author

Reviewer #2: (No Response)

7. PLOS authors have the option to publish the peer review history of their article (what does this mean?). If published, this will include your full peer review and any attached files.

Reviewer #2: No

---

## [Editor Report · Acceptance letter]

17 Oct 2022

PONE-D-21-39757R1 

Psychological Impact, Coping Behaviors, and Traumatic Stress among Healthcare Workers during COVID-19 in Taiwan: An Early Stage Experience 

Dear Dr. Yeh:

I'm pleased to inform you that your manuscript has been deemed suitable for publication in PLOS ONE. Congratulations! Your manuscript is now with our production department. 

Kind regards, 

on behalf of

Dr. Mila Nu Nu Htay 

Guest Editor

PLOS ONE